# HiLoRL: A Hierarchical Logical Model for Learning Composite Tasks

## Abstract

We propose HiLoRL, a hierarchical model to learn policies for composite tasks. Recent studies mostly focus on using human-specified logical specifications, which is laborious and produces models that perform poorly when facing tasks not entirely human-predictable. HiLoRL is composed of a high-level logical planner and low-level action policies. It initially learns a rough rule at its upper level with the help of low-level policies and then uses joint training with surrogate rewards to refine the rough rule and low-level policies. Furthermore, HiLoRL can incorporate specialized predicates derived from expert knowledge, thereby enhancing its training speed and performance. We also design a synthesis algorithm to illustrate our high-level planner's logical structure as an automaton, demonstrating our model's interpretability. HiLoRL outperforms state-of-the-art baselines in several benchmarks with continuous state and action spaces. Additionally, HiLoRL does not require human to hard-code logical structures, so it can solve logically uncertain tasks.

## 1 Introduction

Reinforcement Learning (RL) has achieved tremendous success in a variety of control tasks including robotics (Polydoros, 2017), autonomous driving, and gaming (Mnih, 2015). Despite those achievements, RL algorithms rely heavily on the reward. Some decision-making problems have sparse rewards, making it hard to create a suitable reward function (Ladosz et al., 2022). Rewards must be manually composed for sub-tasks. A well-known example is the OpenAI Fetch Environment (Plappert et al., 2018), in which conventional RL algorithms struggle to learn effective strategies, because of the complexity of reward shaping (Mnih, 2015).

To address the challenges posed by sparse rewards, researchers have explored various strategies. One promising approach that has been proven effective is the use of hierarchical models, aiming to break down complex tasks into manageable sub-tasks (Zhang et al., 2021; Zheng et al., 2022). Those hierarchical models can be broadly categorized into human-provided and non-human-provided (Yu et al., 2023). The human-provided models first determine the upper-level structure by the logical order of sub-tasks and then train the low-level controllers to handle the sub-tasks. This approach simplifies tasks that usually require a lot of exploration and reward shaping for conventional RL models (Hasanbeig & Kroening, 2021). However, they usually come with predefined and fixed high-level planners, leading to a need for deterministic sub-task execution sequences. This greatly limits the performance of the model on some *logically uncertain tasks*, such as autonomous driving on the highway (Leurent, 2018) and card games Hanabi (Bard et al., 2020), where the number and execution sequence of all sub-tasks may vary in every execution of the whole task. In non-human-provided models, purely neural methods are often used for both the upper and lower levels during training (Li et al., 2019). However, this approach may sacrifice the interpretability that logic representation can provide. The interpretability is essential for successful application in high-stakes domains like autonomous driving (Song et al., 2022).

In this paper, we introduce the Hierarchical Logical Reinforcement Learning (HiLoRL) approach. Unlike conventional methods that rely on fixed logical specifications for high-level planning, HiLoRL employs an adaptive logical model as the high-level planner. The logical model has the capability of learning the decision-making process rather than the specific execution sequence, so it can adapt to logically uncertain tasks. A distinguishing feature of our approach is its use of logical

combinations of predicates to capture crucial runtime states, instead of continuous features typical in purely neural methods. This representation not only enhances the model's interpretability by allowing for an automaton summary but also integrates both environmental and expert-designed predicates. The inclusion of these *expert predicates* introduces valuable insights, optimizing training outcomes and expediting the learning process.

Our contributions can be summarized as follows:

• **Adaptive.** Our novel hierarchical reinforcement learning model features an adaptive logical planner. It autonomously learns high-level logical representations, enhancing decision-making and allowing it to tackle logically uncertain tasks effectively.

• **Interpretable.** An automaton is synthesized from our logical planner, using predicate representation for states and decisions for transitions. This results in a human-readable structure, leveraging the inherent interpretability of predicate representations.

• **Instructable.** Our model is designed to optionally integrate expert knowledge. While it can operate autonomously, the incorporation of domain expertise further enhances its training efficiency and results.

## 2 PRELIMINARIES

### 2.1 MARKOV DECISION PROCESS

The Markov Decision Process (MDP) (Puterman, 1990) is a mathematical model that represents decision-making under uncertainty. It consists of a tuple $(S, A, P, R, \gamma)$, where $S$ denotes states, $A$ represents actions, $P$ is the transition function, $R$ is the reward function, and $\gamma$ is the discount factor. Given the current state $s_t$, an agent picks action $a_t$ leading to a new state $s_{t+1} \sim P(\cdot|s_t, a_t)$ and obtains reward $r_{t+1} = R(s_t, a_t, s_{t+1})$. The goal of an MDP is to identify an optimal policy $\pi$ maximizing the expected cumulative reward over time. This is realized by determining the value function $V(s)$, which reflects the expected discounted reward from state $s$ using policy $\pi$.

### 2.2 FIRST ORDER LOGIC

First-order logic (FOL) (Barwise, 1977) is a formal language used to describe objects in the world and relations among them. FOL typically consists of several elements: constants, variables, predicates, and clauses. Constants typically correspond to specific objects in the environment, while variables represent unspecified constants. Predicates can be denoted by the symbol $P$, and an n-ary predicate is denoted as $P(x_1, x_2, ..., x_n)$, where $x$ represents constants or variables. Predicates commonly represent the properties of objects or relationships between objects. The value of predicates can be either true or false. Clause is a rule in the form $p_1 \leftarrow p_2, p_3, ..., p_n$, where $p_1$ is the head atom and $p_2, p_3, ..., p_n$ are body atoms. Predicates formed by constants are referred to as extensional predicates, which are used for input predicates in our model. Predicates formed by a series of clauses are known as intensional predicates, which are used for target predicates in our model.

### 2.3 DIFFERENTIABLE LOGIC MACHINE

Differentiable Logic Machine (DLM) (Zimmer et al., 2021) is a trainable model proposed for doing calculations between predicates. It takes in a series of predicates as input and combines different predicates using the basic logic relation: fuzzy and $\wedge$, fuzzy or $\vee$, and fuzzy not $\neg$.

DLM has a max depth of $D$. In each layer, it has $B$ computing units which are for predicates of different element numbers. We denote the $b$-ary predicates in the d th layer in computing unit $b, b \in \{0, 1, ..., B-1\}$, e.g. $(P_d^b(x_1, x_2, ...x_b))$. The unit $b$ output a $b$-ary predicate after computation and pass it to the next layer. Apart from computing using predicates of the same argument number, DLM has expanded and reduced operations to enable computation between predicates of different argument numbers. For computing unit $b$, it can get predicates from the former layer. The compute operation can be formed using binary operation $P_d^b = P_{d-1}^x \wedge P_{d-1}^y$, $P_d^b = P_{d-1}^x \vee P_{d-1}^y$, $P_d^b = P_{d-1}^x \wedge \neg P_{d-1}^y$, $P_d^b = P_{d-1}^x \vee \neg P_{d-1}^y$, where $x, y \in \{b-1, b, b+1\}$.

# 3 HIERARCHICAL DIFFERENTIABLE LOGIC REINFORCEMENT LEARNING

## 3.1 MODEL FRAMEWORK

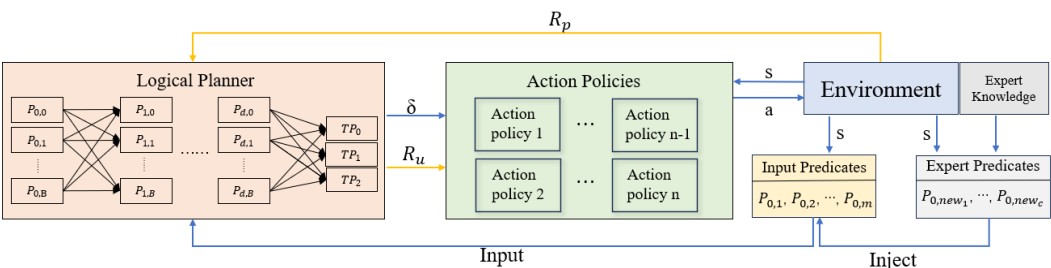

Figure 1: Model framework. ($R_p$ refers to the rewards obtained from the environment, $R_u$ refers to the rewards feedback from the logic planner, which will be introduced in Section 3.2.2, $s$ refers to the state space, $a$ refers to the action space, $\delta$ is the output of the logical planner as the decision to choose the specific action policy, which is introduced in Equation 2).

We propose a hierarchical model for planning policies for MDPs. Roughly speaking, this model makes the assumption that an integrated task can be divided into a number of natural and logically interdependent sub-tasks. Based on this assumption, our model incorporates two levels: the *high-level logical model* for sub-task identification and planning, and at the lower level, there is a pool of the *low-level action policies* that are responsible for accomplishing a variety of sub-tasks. More specifically, an *input module* is responsible for translating the MDP states into *input predicates*. The input predicates are then fed into the upper-level model which is realized by a DLM. The DLM, at its output layer, makes a decision on the choice of a low-level action policy. The chosen low-level policy, taking the MDP states as input, makes a final decision on which MDP action to take. The whole decision process is illustrated in Figure 1, and formally described as follows.

**Input module and predicates.** As mentioned above, the input module transforms the MDP observations $s_t \in \mathbb{R}^K$ (at time $t$) into logical predicates that serve as the inputs of the high-level DLM. In general, a $b$-ary logical predicate $P$ is defined based on a $b$-ary real *transformation function* $f : \mathbb{R}^b \to \mathbb{R}$, a list of indices $i_1, i_2, \ldots, i_b \in [K]$, and an *activation interval* $(u, v)$. The predicate is then generated by

$$P(s_t) \leftarrow f((s_t)_{i_1}, (s_t)_{i_2}, \ldots, (s_t)_{i_b}) \in (u, v).$$

The arity of a usual logical predicate used in our model is at most 3, which is enough for our experiments. We adopt the transformation functions in the simple forms of addition/subtraction so that they can be generally useful for most natural tasks. We list the adopted functions in the following table and note that functions such as $(x, y, z) \mapsto x+y-z$ can be substituted by $(x, y, z) \mapsto x-y+z$ via changing the argument order.

| Arity | Transformation functions |
|---|---|
| 1 | $x \mapsto x$ |
| 2 | $(x, y) \mapsto x + y, \ (x, y) \mapsto x - y$ |
| 3 | $(x, y, z) \mapsto x + y + z, \ (x, y, z) \mapsto x + y - z$ |

Going through the combinations of the transformation functions, the indices, and the activation intervals, the input module obtains a sequence of predicates and forms the set of input predicates.

$$\mathcal{P}_0 = \{P_{0,1}, P_{0,2}, P_{0,3}, \ldots, P_{0,m_0}\} \tag{1}$$

**Injecting expert knowledge via special predicates.** In the input module, one may introduce predicates with special transformation functions on particular indices of the MDP state vector, based on the expert understanding of a specific task. This can help to substantially improve the planning performance. Please refer to Section 4.4 for detailed demonstrations.

**High-level decision and the choice of an action policy.** Suppose there are $n$ low-level action policies. After the final layer (layer $d$) of the high-level DLM, we append a fully connected layer (layer $(d + 1)$) so that there are $n$ special *target predicates*, $\text{TP}_1, \text{TP}_2, \ldots, \text{TP}_n$, each of which corresponds to an action policy. An index $\delta_t$ is sampled via

$$\delta_t \sim \text{softmax}\{\text{TP}_1, \text{TP}_2, \ldots, \text{TP}_n\}. \tag{2}$$

**Deciding the MDP action.** Finally, we invoke the low-level action policy $\pi_\delta$ and take the MDP action.

$$a_t \leftarrow \pi_{\delta_t}(s_t) \tag{3}$$

**Focused predicate extraction.** We adapt the output of DLM so that it can output critical predicate after training. As is mentioned in Section 2.3, DLM do the logical computation in every depth using the equation $P_d = (\sum w_{p_{d-1}} P_{d-1}) \odot (\sum w_{p'_{d-1}} P'_{d-1})$. $\odot \in \{\wedge, \vee, \wedge\neg, \vee\neg\}$. Since this is a binary computation, we try to extract the predicate $P_{d-1}$ and $P'_{d-1}$ with the largest $w_{p_{d-1}}$ and $w_{p'_{d-1}}$. We perform this operation from the last layer $d$ to the first layer, decomposing the predicate with 2 predicates of the largest weight in the former layer. We define the extracted predicates in the input layer as the *focused input predicate* $P^*$. This new functionality greatly contributes to our interpretability experiment in Section 4.3 regarding automaton representation.

## 3.2 Training

The low-level action policies are expected to deal with relatively simple sub-tasks that commonly arise in many planning and control problems. Following the previous works (Jothimurugan et al., 2021; Yang et al., 2020), we assume access to a pool of pre-selected and pre-trained action policies. On the other hand, we note that the conceptually "same" sub-task may have delicate differences, and appropriately addressing these differences might be crucial to the performance of the entire hierarchical model. For example, in the gym fetch environment (Plappert et al., 2018), the *PickAndPlace* tasks might involve a "same" sub-task named *approach* — where the robotic arm is maneuvered closer to an object. This movement becomes particularly crucial when the size of the object undergoes a change, affecting the arm's subsequent ability to pick up the object. The nuances in these main tasks mean the *approach* sub-task must be adjusted based on the object's size. Consequently, there's an evident need to refine these pre-trained low-level action policies to ensure they're optimally attuned to the primary task.

In light of the above discussion, we assume the pre-trained low-level action policies $\pi_i(\cdot) = \pi_i(\cdot|\theta_i)$ can be tuned further by optimizing their parameters $\theta_i$ via policy gradient methods. In the rest of this section, we describe how to separately train the high-level DLM and the low-level action policies while fixing the other part, as well as jointly train the two parts. A highlight of our training algorithms is the design of the surrogate rewards and the training objectives, which crucially rely on our model structure and help to achieve superior model performance.

### 3.2.1 Training the High-level Logical Model

Here we illustrate how to train the high-level logical model based on the environment rewards $\{r_t\}$ while fixing the low-level action policies. It is well-known that the natural environment rewards in control problems are usually sparse (e.g., there is only a positive reward when the task is successfully done), and this is challenging for RL algorithms (Mnih et al., 2013). Our hierarchical model offers a different approach to address this challenge – whenever the high-level logical model decides the choice of a low-level action policy $\pi_\delta$, we invoke $\pi_\delta$ for a number of consecutive time periods (namely a *volley*) instead of only once, we aggregate the environmental rewards obtained in the volley and train the high-level logical model using these *volley rewards*. In Algorithm 1, we roll-out the trajectory based on volleys: $\{(s'_v, \delta'_v, r'_v)\}_{v \in \{0,1,2,\ldots\}}$. These volley-based trajectories are shorter and the volley rewards become less sparse, which is helpful to the RL algorithms.

We apply the standard PPO algorithm (Schulman et al., 2017) to the volley-based trajectory $\{(s'_v, \delta'_v, r'_v)\}_{v \in \{0,1,2,\ldots\}}$ to optimize the high-level policy $\pi(\cdot|\theta_{\text{DLM}})$. In PPO, we also train a neural network fed by the input predicates as the function approximated to the critic functions.

**Algorithm 1:** Volley-based Roll-out for High-level Logical Model Training

**Input:** high-level DLM policy $\pi(\cdot|\theta_{\text{DLM}})$ as described in Eq. (2), low-level action policies $\{\pi_i(\cdot|\theta_i)\}$, horizon $T$, volley size $\tau_{\text{volley}}$

1   Volley count $v \leftarrow 0$;
2   **while** $v < T/\tau_{\text{volley}}$ **do**
3      observe the environment state $s_{v \cdot \tau_{\text{volley}}}$, let $s'_v \leftarrow s_{v \cdot \tau_{\text{volley}}}$;
4      calculate the input predicates $\mathcal{P}_0$ based on $s'_v$, sample an index $\delta'_v \sim \pi(s'_v|\theta_{\text{DLM}})$;
5      Volley reward $r'_v \leftarrow 0$;
6      **for** $j \leftarrow 0$ **to** $\tau_{\text{volley}}$ **do**
7         if $j \neq 0$ then observe the environment state $s_{v \cdot \tau_{\text{volley}}+j}$;
8         execute the environment action $a_{v \cdot \tau_{\text{volley}}+j} \leftarrow \pi_{\delta'_v}(s_{v \cdot \tau_{\text{volley}}+j}|\theta_{\delta'_v})$, receive the environment reward $r_{v \cdot \tau_{\text{volley}}+j}$;
9         $r'_v \leftarrow r'_v + r_{v \cdot \tau_{\text{volley}}+j}$;
10      **end**
11      $v \leftarrow v + 1$;
12   **end**

**Algorithm 2:** DDPG-based Roll-out for Low-level Action Policy Training

**Input:** high-level DLM policy $\pi(\cdot|\theta_{\text{DLM}})$ as described in Eq. (2), low-level action policies $\{\pi_i(\cdot|\theta_i)\}$, low-level critic $\{Q_i(\cdot|\theta_{Q_i})\}$, horizon $T$, volley size $\tau_{\text{volley}}$, learning rate for actor $\alpha$, learning rate for critic $\beta$

1   $t \leftarrow 0$, observe the environment state $s_0$;
2   **while** *task not completed and $t < T$* **do**
3      calculate the input predicates $\mathcal{P}_0$ based on $s_t$;
4      sample an index $\delta_t \sim \pi(s_t|\theta_{\text{DLM}})$;
5      **for** $j \leftarrow 1$ **to** $\tau_{\text{volley}}$ **do**
6         obtain action $a_{t+j}$ from $\pi_{\delta_t}(s_{t+j}|\theta_{\delta_t})$;
7         receive the environment reward $r_{t+j}$, observe the new environment state $s_{t+j+1}$;
8         get the estimated value $\omega_{t+j}$ from the critic network of DLM;
9         $\theta_{\delta_t} \leftarrow \nabla(\log \pi_{\theta_{\delta_t}}(a'_v|s'_v)Q_{\delta_t}(s_{t+j-1}, a_{t+j-1})) + \alpha \cdot \theta_{\delta_t}$;
10         $\theta_{Q_{\delta_t}} \leftarrow \nabla(R + \gamma Q_{\delta_t}(s_{t+j}, a_{t+j}) - Q_{\delta_t}(s_{t+j-1}, a_{t+j-1})) + \beta \cdot \theta_{Q_{\delta_t}}$;
11      **end**
12      $t \leftarrow t + \tau_{\text{volley}}$;
13   **end**

### 3.2.2 TRAINING THE LOW-LEVEL ACTION POLICIES

Now we fix the high-level DLM and train the low-level action policies using the policy gradient RL algorithm. In principle, one may use most off-the-shelf RL algorithms in this step, such as Deep Deterministic Policy Gradient (DDPG) Lillicrap et al. (2015).

During training, we roll out the trajectory for the whole task, and the pieces of the trajectory are sorted and collected according to the selected low-level action policy. Finally, we train each low-level action policy separately by gradient descent using the collected data.

The (DDPG-based) roll-out algorithm is described in Algorithm 2. The key is at Line 8, where we use a surrogate reward defined based on the following function (where $\alpha > 0$ is a hyper-parameter):

$$R(r, \delta, \delta', \omega, \omega') = r + \alpha \times 1[\delta \neq \delta' \wedge \omega > \omega']. \tag{4}$$

The idea of Eq. (4) is that we combine the environment reward $r$, the instruction from the high-level logical model, and the estimated value $\omega$. $\omega$ is the value function in RL, which is the output of the DLM critic network. When the current sub-task is completed, the high-level model would choose a different action policy so that $\delta \neq \delta'$ and the expectation of value also increases $\omega > \omega'$. Therefore, adding the term $\alpha \times 1[\delta \neq \delta' \wedge \omega > \omega']$ incentivizes the low-level action policy to learn to complete the sub-task requested by the high-level model.

### 3.2.3 JOINT TRAINING

In our holistic training approach, we harmoniously employ both Algorithm 1 and Algorithm 2. For each epoch, we gather trajectories for both the high-level logical planner and the low-level action policies, encapsulating the transitions of these layered policies within a single roll-out. Crucially, at every epoch, parameters for both the high-level and low-level models are diligently updated, ensuring a dynamic and adaptive learning environment.

This integrated and consistent approach fosters a symbiotic relationship between high-level and low-level policies. By training them in tandem, and ensuring both are iteratively refined at each epoch, their collaborative efficiency is enhanced. Each layer informs and refines the other, driving both towards more seamless cooperation and, consequently, a jointly optimal solution. Through this

method, our entire system benefits from the synergistic evolution of its components, achieving great adaptability and performance.

## 4 EXPERIMENT

We evaluate our approach on two sparse reward scenarios, *highway environment* and *fetch environment* focusing on performance and functionality. This section is organized as follows: In Sections 4.1 and 4.2, we demonstrate the performance of our model. Sections 4.3 and 4.4 delve into the main features of HiLoRL, emphasizing its interpretability and the integration of expert knowledge. In Section 4.5, we display the fine-tuning of low-level action policies using Algorithm 2. Some additional details of our experiment are shown in the Appendix E.

### 4.1 HIGHWAY ENVIRONMENT

The highway environment is an autonomous driving simulator based on the OpenAI Gym Library (Leurent, 2018). The goal of the task is to control a car agent to maintain a high speed and avoid crashing at the same time. The state space is $\mathbb{R}^{25}$, which describes the states of the ego agent and the four closest vehicles to it. Each vehicle is described by 5 features: a binary existence flag, the x-axis position, the y-axis position, the x-axis velocity, and the y-axis velocity. The action space is $\mathbb{R}^2$, which describes the horizontal acceleration and angular acceleration of the ego agent.

**Results**: The agents are tested with a maximum of 100 time steps in each episode. The performance comparison is shown in Table 1. The *crash rate* is the percentage of episodes that end with a crash. The *velocity* is the average velocity of the agent. The *length* measures the average duration the agent car remains active in the scenario. We evaluate the performance of some conventional continuous RL algorithms (SAC, PPO). We find that they have a poor performance compared with HiLoRL. In SAC and PPO, the agent takes a negative policy to decelerate from the very beginning to avoid the collision. Notably, HiLoRL* represents our model without the benefits of joint training, and the results show a significant increase in performance after joint training.

| | SAC | PPO | HiLoRL | HiLoRL* |
|---|---|---|---|---|
| length | 100 | 100 | 96.42 | 95.53 |
| velocity | 10.46 | 12.43 | 26.65 | 25.70 |
| crash rate | 0% | 0% | 4.9% | 5.4% |

| | PPO$^{\triangle}$ | NLM$^{\triangle}$ | DLM$^{\triangle}$ | HiLoRL |
|---|---|---|---|---|
| length | 93.43 | 95.8 | 96.77 | 96.42 |
| velocity | 22.22 | 25.20 | 26.36 | 26.65 |
| crash rate | 7.2% | 5.4% | 4.3% | 4.9% |

Table 1: Performance comparison for highway using continuous control. HiLoRL* refers to the model without joint training.

Table 2: Performance comparison with easier discrete control mode (The model marked with the superscript $\triangle$ uses discrete control mode).

Since the performance of the past model on *continuous control* scenario is relatively poor, we also compare HiLoRL with some *simpler discrete control* methods in Table 2. Remarkably, even when operating in the more demanding continuous control environment, HiLoRL achieves performance comparable to that of DLM and surpasses both PPO and NLM. It is noteworthy to mention that while HiLoRL thrives in a continuous setting, which is inherently more complex, the original DLM and NLM are not even designed to support continuous control scenarios. This reflects HiLoRL's superiority in managing more intricate control challenges.

### 4.2 FETCH ENVIRONMENT

The *Fetch-Pick-And-Place* environment in OpenAI Gym consists of a robotic arm and an object (Plappert et al., 2018). The goal of the task is to pick the object and place it in a certain position. The state space is $\mathbb{R}^{25}$, including the position of the gripper, the position of the object, and the distance between gripper fingers. These three types of components are the main components we focus on. The action space is $\mathbb{R}^4$. The first three components encode the target gripper position and the last component represents the target gripper width. The initial position of the object on the table is randomly generalized.

In our experiment, we design three complicated tasks for our model. *Pick&Place*: Let the robot arm grab the object to a target position. *Pick&PlaceCorner*: Let the robot arm grab the object, then lift

it, and finally grab it to the top right corner. *PickLiftPlace*: Let the robot arm grab the object, then lift it, and finally grab it to the target position.

| Succ rate | QRM | SPECTRL | DiRL | HiLoRL | HiLoRL* |
|---|---|---|---|---|---|
| Pick&Place | 0.00 | 0.00 | 93.33 | **95.67** | 92.67 |
| Pick&PlaceCorner | 0.00 | 0.00 | 93.33 | **95.33** | 94.67 |
| PickLiftPlace | 0.00 | 0.00 | 92.00 | **95.00** | 91.67 |

Table 3: Performance comparison for fetch, HiLoRL* refers to the model without joint training.

**Results**: The performance of different models in different fetch tasks is listed in Table 3. We use some state-of-the-art logical specification models: DiRL (Jothimurugan et al., 2021), SPECTRL (Kishor Jothimurugan & Bastani, 2019), and QRM (Rodrigo Toro Icarte & McIlraith., 2018) as baseline. QRM algorithm performs poorly because it lacks the procedure of decomposing the problem into simpler sub-problems and they do not integrate model-based planning at the high level.

For baseline DiRL, we build in the sequential relation of sub-tasks using its specification. HiLoRL and DiRL both adopt stateful policies that are effective for sparse reward tasks, so these two models can achieve a success rate of over 90%. Compared with DiRL, in which the high-level logic of the whole task is built in advance, our model generates the high-level logic during task execution. HiLoRL is equipped with self-adaptive decision-making to handle unexpected situations, like when a low-level action policy fails to grab an object. In such cases, the high-level planner can revert to the initial policy and re-start the task from scratch. Such kind of feature is not found in task decomposition with predefined logic specification decomposition(such as DiRL). This adaptiveness ensures a superior performance over DiRL.

We also highlight the success rate of HiLoRL when executed *without joint training*. The substantial enhancement in the success rate, when utilizing our joint training algorithm, underscores its effectiveness. This improvement stems from the joint training algorithm's ability to simultaneously refine both the low-level action policies and the high-level logical planner, allowing for a more harmonious cooperation between the two levels.

### 4.3 AUTOMATON REPRESENTATION

Different from the works that use formal language to define their high-level logic (Jothimurugan et al., 2021), our high-level model can learn more complicated logical structures that can be summarized as automaton representations. In this part, we show how to synthesize a Deterministic Finite Automaton (DFA) from our high-level planner and evaluate the correctness of the DFA. The detailed algorithm is depicted in Appendix A.

**First of all**, we extract the *focused input predicate* after training. This approach allows us to focus on the input predicates that truly impact the decision-making process and groups the observations from the environment into a limited number of automaton states. **Secondly**, we utilize the HiLoRL to track changes in the high-level planner's decision. We run the HiLoRL and record the true value of the focused input predicate $P^*$ and the new decision $\delta$ every time the high-level planner's output decision changes. The $P^*$ and $\delta$ are defined as the state and transition edge of the DFA. With this approach, each run yields a path in the automaton. Then we merge the nodes with the same $P^*$ and the edges with the same $\delta$ and prior node to get a complicated automaton. **Finally**, we apply the Hopcroft Algorithm (Gries, 1973) to simplify the automaton.

We present the reduced automaton for an ego agent to overtake and return to its original lane when there is a car ahead and no other cars in the environment. As we depict in Figure 2, this case is decomposed into 7 logical states. The final state is a terminate state, which means the overtaking process is finished. Meanwhile, we also provide the focused predicate representation for each state, which is shown on the right of Figure 2. The predicate representation of each state shows the important information that our logical planner focuses on. For example, if we take the "accelerate" action and there are no vehicles around us in our target lane to the right(i.e. the state $q_4$), the model will decide to "merge right" in the next step until it achieves the state $q_6$. It is demonstrated that

HiLoRL can create easy-to-understand automata with logical explanations for each state, improving interpretability. The complete version of the focused predicate representation for each state can be found in Appendix B. The automaton before the reduction step is shown in the Appendix C.

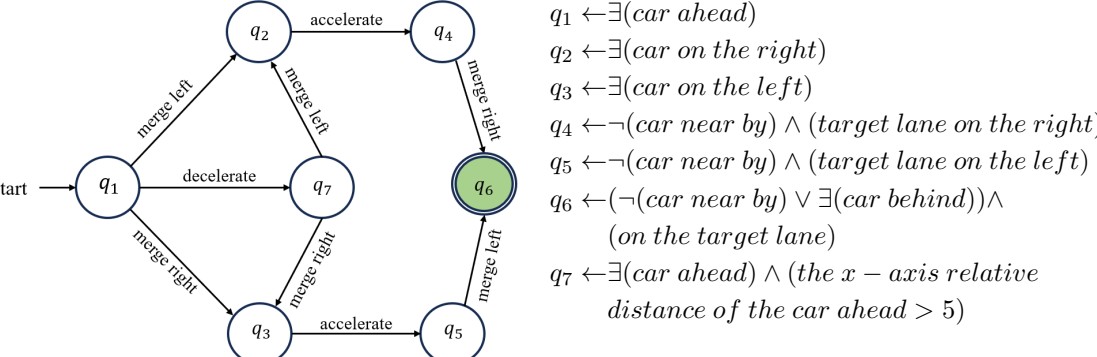

Figure 2: (Left) Automaton representation for overtaking task. The edge of the automaton represents the low-level action policy. (Right) The description of each automaton state is summarized from the focused input predicates.

## 4.4 EXPERT KNOWLEDGE INSTRUCTION

Compared with other traditional RL methods, the use of logical predicates as inputs in our high-level planner makes it possible to inject *expert knowledge* into the model. We can leverage expert knowledge to design specific functions based on human awareness and understanding of a certain task. Utilizing these expert predicates as inputs to the high-level planner indeed enhances the training effectiveness and efficiency of HiLoRL.

Here we conduct a comparative experiment. We compare the success rate of the tasks using different input predicates. For the basic model, the transformation functions are in the simple forms of addition and subtraction (listed in Section 3.1). Contrarily, we summarize some human-defined predicates for input with expert knowledge, which is the norm 1 distance between the object and the gripper. Namely speaking, we compose 4 predicates in the original input: $P_{0,1} = x_{object} - x_{gripper}, P_{0,2} = y_{object} - y_{gripper}, P_{0,3} = z_{object} - z_{gripper}, P_{0,4} = g_{left} - g_{right} - t$ together to generate *an expert predicate*:

$$P_{0,new} = |x_{object} - x_{gripper}| + |y_{object} - y_{gripper}| + |z_{object} - z_{gripper}| + |g_{left} + g_{right} - t| \leq err \tag{5}$$

$t$ is a threshold for the distance between 2 grippers, $g_{left}$ and $g_{right}$ is the displacement of left and right gripper. With different $t$ values, $P_{0,new}$ can judge whether the object is grabbed by the grippers and whether the gripper is able to grab the object. $err$ is a tolerable error range.

| Succ rate | HiLoRL | HiLoRL(DK) |
|---|---|---|
| Pick&Place | 95.67 | **97.33** |
| Pick&PlaceCorner | 95.33 | **99.33** |
| PickLiftPlace | 95.00 | **99.00** |
| Epoch | 149.00 | **60.00** |

| Succ rate | Pre-Finetune | Post-Finetune |
|---|---|---|
| Pick&Place | 84.33 | **94.00** |
| Pick&PlaceCorner | 85.33 | **95.67** |
| PickLiftPlace | 84.00 | **93.00** |

Table 4: Performance comparison for domain knowledge instruction (DK means the HiLoRL with domain knowledge instruction).

Table 5: Performance comparison before and after fine-tuning after changing the object size.

**Result:** The performance comparison is listed in Table 4. The failure rate of HiLoRL is reduced by more than $50\%$ with expert knowledge. This is because we have more precise predicates for the high-level planner. We reduce the complexity of the relation that needs to be learned by the upper-level planner by building in a part of the relation. Additionally, we also compare the number of epochs for convergence and find that HiLoRL needs fewer epochs with expert knowledge.

### 4.5 MODEL FINE-TUNING

To show the effectiveness of our fine-tuning algorithm in tasks that are conceptually analogous yet distinct in their details, we make modifications to the environment, reducing the side length of the cubic object from 0.25cm to 0.15cm. It is worth noting that we only *fine-tune a certain set of lower-level action policies* using Algorithm 2 while keeping the high-level logical planner unchanged.

**Result:** The experiment results are depicted in Table 5. After adjusting the size of the object, the success rate of the model decreases by approximately 10%. However, through solely fine-tuning the lower-level action policies, we are able to effectively recover the success rate lost. Additionally, since the lower-level action policy is relatively simple and has a low training cost, it signifies that we can quickly fine-tune our model to adapt to changes in the environment and task requirements.

## 5 RELATED WORK

Recent advancements in Knowledge Representation and Reasoning (KRR) have seen models employing logical languages for efficient abstract representation within the RL framework. Notably, Icarte et al. (2018) utilizes finite automata for high-level task decomposition, introducing reward machines as a superior alternative to manual reward functions. Given the limitations of manually designing rewards in non-Markovian scenarios, there's been a significant shift toward using Temporal Logic (TL). For instance, Brafman et al. (2018) uses LDLf for specifying non-Markovian rewards with corresponding automata constructions, while Jothimurugan et al. (2019) offers the SPECTRL specification language for encoding complex task sequences. Despite their efficacy in challenging domains, a common drawback is their reliance on explicit prior knowledge, restricting their use in ambiguous tasks (Yu et al., 2023). Another promising approach integrates symbolic planning with RL: Yang et al. (2018) presents the PEORL framework, blending symbolic planning with hierarchical RL, and Illanes et al. (2020) explores symbolic action models combined with model-free RL. Closely aligned with our work, DiRL (Jothimurugan et al., 2021) introduces an algorithm splitting policy synthesis into separate planning and control problems.

Inductive logic programming (ILP) (Lavrac & Dzeroski, 1994) refers to the problems of extracting effective rules from a limited set of rule templates. Evans & Grefenstette (2018) introduces differentiable operations into the training framework of ILP and develops a novel ILP model that can be trained using gradient descent which achieves great success on various benchmarks. The major challenge in the ILP field lies in its difficulty in scaling up to complex scenarios (Cropper et al., 2022). As the complexity of the tasks increases, the search space of rules grows exponentially. Dong et al. (2019) proposes a neural logic machine(NLM) that resembles a neural network, where parameters are differentiable. By introducing MLP into the model training process, the model's capacity is greatly enhanced. However, this enhancement comes at the cost of interpretability. Building upon the foundation of NLM, Zimmer et al. (2021) proposes a more interpretable model DLM. This model introduces fuzzy logic operations in the framework, replacing the computation process of MLP. Jiang & Luo (2019) first introduces the application of ILP methods in the RL tasks. It leverages the principles of $\delta$ILP (Evans & Grefenstette, 2018) and combines them with the training framework of the PPO (Schulman et al., 2017) algorithm. This approach has achieved success in various reinforcement learning scenarios such as Blocksworld and Cliff-Walking (Jiang & Luo, 2019).

## 6 CONCLUSION

We introduce HiLoRL, an adaptive hierarchical logical reinforcement learning model. It integrates a logical planner for high-level decision-making and action policies designed for the precise execution of sub-tasks. Notably, HiLoRL excels in continuous control tasks and stands out due to its interpretability and instructability. Moreover, it has demonstrated a swift adaptability to changes in the environment. For future work, we aim to construct a hierarchical system with minimal domain knowledge, which offers a promising avenue toward establishing a lifelong learning system.

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

## A  DFA Synthesize Algorithm

Here we provide a detailed description of our automaton synthesize program. The output of the high-level logical model is a possibility distribution for each low-level action policy. We select the action policy $\pi_\delta$ based on this possibility distribution. We invoke the corresponding policy $\pi_\delta$ also for a number of consecutive periods. During this process, we track the value of the *focused predicate*. We define all the observation $s$ with the same *focused predicate* as a new state $s_{auto}$ for the automaton. We group the state $s_{auto_v}$ with the same predecessor automaton state $s_{auto_{v-1}}$ and transition $\delta_{v-1}$ into a collection, which is represented by a new state $S_i$ in the automaton. This process is repeated until the end of the episode.

In the experiment, we collect a large amount of traces (specifically 100,000) and group the observations into different automaton states. Additionally, we employ a predicate for judging whether the task is accomplished, so that we can easily figure out the terminate state for the automaton. Finally we apply the Hopcroft Algorithm to simplify the automaton.

The pseudo-code for this algorithm is Algorithm 3.

## B  Focused Predicates for Automaton of Overtaking Task

In this section, we detail the predicate representation for states $q_i$ as illustrated in Equation 2. The predicate representations of the automaton states can be seen in Equation 6. The agent is regarded as in state $q_i$ when the logical expression for $q_i$ holds true.

$$
\begin{aligned}
q_1 &\leftarrow P_{0,1} \land \neg P_{0,2} \land \neg P_{0,3} \land \neg P_{0,4} \\
q_2 &\leftarrow \neg P_{0,1} \land \neg P_{0,2} \land \neg P_{0,3} \land P_{0,4} \\
q_3 &\leftarrow \neg P_{0,1}, \land \neg P_{0,2} \land P_{0,3} \land \neg P_{0,4} \\
q_4 &\leftarrow \neg P_{0,1} \land \neg P_{0,2} \land \neg P_{0,3} \land \neg P_{0,4} \land P_{0,85} \\
q_5 &\leftarrow \neg P_{0,1} \land \neg P_{0,2} \land \neg P_{0,3} \land \neg P_{0,4} \land P_{0,87} \\
q_6 &\leftarrow \neg P_{0,1} \land \neg P_{0,3} \land \neg P_{0,4} \land P_{0,86} \\
q_7 &\leftarrow P_{0,1} \land \neg P_{0,2} \land \neg P_{0,3} \land \neg P_{0,4} \land (P_{0,8} \lor P_{0,9})
\end{aligned}
\tag{6}
$$

Here we list the focused input predicates mentioned in Equation 6, and they are all contained in Table 8.

$P_{0,1}$:  if there is a car ahead.
$P_{0,2}$:  if there is a car behind.
$P_{0,3}$:  if there is a car on the left.
$P_{0,4}$:  if there is a car on the right.
$P_{0,85}$:  if ego car is on the left of the target lane.
$P_{0,86}$:  if ego car is on the target lane.
$P_{0,87}$:  if ego car is on the right of the target lane.
$P_{0,8}$:  if the x-axis relative distance of the car ahead and ego car is between 5 and 10.
$P_{0,9}$:  if the x-axis relative distance of the car ahead and ego car is larger than 10.

From Figure 2 and Equation 6 we can give a description for each automaton state in the overtaking task. We start from $q_1$, where there is a car in front of the ego agent. Then the ego agent can take three feasible actions: *decelerate*, *merge left*, *merge right*. If the ego agent chooses to decelerate, it will reach $q_7$. $q_7$ and $q_1$ are almost the same except the distances between two cars become larger. If it takes left (right) lane change action, we can find the value-focused predicate $P_{0,4}(P_{0,3})$ changes. Then the agent accelerates until focused input predicate $P_{0,4}, P_{0,3}$ all become false, which means it is a proper time to get to the origin lane. Finally, it takes right (left) lane change to finish overtaking (reach $q_6$).

---

**Algorithm 3:** Synthesis Automaton Logic Representation for High-level Policy

---

**Input:** high-level DLM policy $\pi(\cdot|\theta_{\text{DLM}})$ as described in Eq. (2), low-level action policies $\{\pi_i(\cdot|\theta_i)\}$, horizon $T$, volley size $\tau_{\text{volley}}$, epoch $N$

1   epoch count $n \leftarrow 0$;
2   automaton state node map $\mathcal{M}$, the key of $\mathcal{M}$ is the state node of automaton, while the value is another submap describing the decision and the corresponding next state;
3   **while** $n < N$ **do**
4      Volley count $v \leftarrow 0$;
5      **while** *task not completed and* $v < T/\tau_{\text{volley}}$ **do**
6          calculate the input predicates $\mathcal{P}_0$ based on $s_v$;
7          sample an index $\delta_v \sim \pi(s_v|\theta_{\text{DLM}})$;
8          **if** $\delta_v \neq \delta_{v-1}$ **then**
9              extract focused input predicates $\mathcal{P}^*$ from $\pi(\cdot|\theta_{\text{DLM}})$ for the output $\delta_v$;
10             calculate the true value of $\mathcal{P}^*$ based on $s_v$;
11             automaton state $S_{auto_v} \leftarrow \mathcal{P}^*$;
12             **if** $S_{auto_{v-1}}$ *in* $\mathcal{M}$ **then**
13                 **if** $(\delta_{v-1}$ *in* $\mathcal{M}[S_{auto_{v-1}}]$ **then**
14                    $S_{exist} \leftarrow \mathcal{M}[S_{auto_{v-1}}][\delta_{v-1}]$;
15                    merge $S_{exist}$ and $S_{auto_v}$ because they represent the same state in automaton;
16                 **end**
17                 **else**
18                    add $\{\delta_{v-1} : S_{auto_v}\}$ to $\mathcal{M}[S_{auto_{v-1}}]$;
19                 **end**
20             **end**
21             **else**
22                 add $\{S_{auto_{v-1}} : \{(\delta_{v-1} : S_{auto_v})\}\}$ to $\mathcal{M}$;
23             **end**
24          **end**
         **for** $j \leftarrow 0$ *to* $\tau_{\text{volley}}$ **do**
            if $j \neq 0$ then observe the environment state $s_{v \cdot \tau_{\text{volley}} + j}$;
            execute the environment action $a_{v \cdot \tau_{\text{volley}} + j} \leftarrow \pi_{\delta_v}(s_{v \cdot \tau_{\text{volley}} + j}|\theta_{\delta_v})$;
         **end**
25      **end**
26      $v \leftarrow v + 1$;
27   **end**
28   Split all nodes into Accept state $A$ (Terminate State) and Non Accept state $N$;
29   $N \leftarrow \{S \backslash TerminateState\}$ ;
30   **while** *True* **do**
31      **for** *each state set $T$ in $N$* **do**
32          **for** *each $\delta$ in option set* **do**
33             **if** *$\delta$ can split $T$* **then**
34                 split $T$ into $T_1 ... T_k$;
35                 add $T_1 ... T_k$ to N;
36             **end**
37          **end**
38      **end**
39      **if** *no split operation is done* **then**
40          break;
41      **end**
42   **end**

---

## C UNSIMPLIFIED AUTOMATON FOR HIGHWAY TASK

The automaton prior to simplification using the Hopcroft algorithm is presented in Figure 3. When we compare this with the simplified version in Figure 2, several state reductions can be observed:

• The terminate states $q_6, q_8, q_{14}, q_{15}$ are grouped into $q_6'$.

• $q_2$ and $q_{10}$ are grouped into $q_2'$.

• $q_4$ and $q_{12}$ are grouped into $q_4'$.

• $q_3$ and $q_{11}$ are grouped into $q_3'$.

• $q_5$ and $q_{13}$ are grouped into $q_5'$.

Notably, all states succeeding $q_7$ have equivalent states in the simplified version, indicating that the "decelerate" operation is non-essential for the agent's overtaking task.

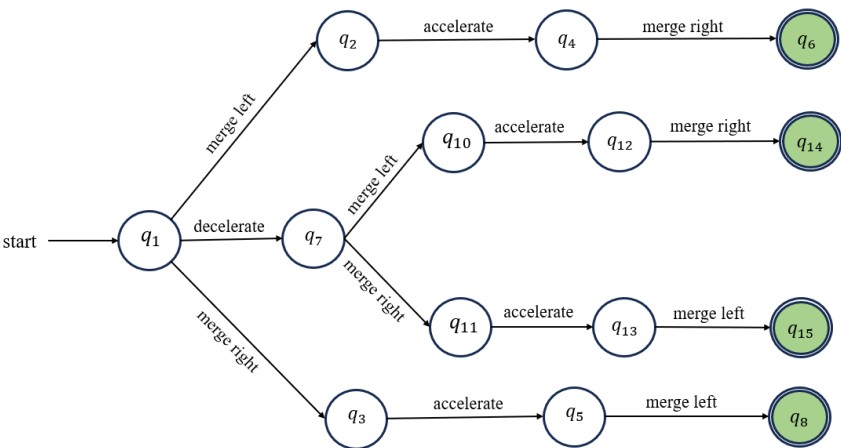

Figure 3: Automaton for highway task before reduction.

## D AUTOMATON REPRESENTATION FOR FETCH TASK

We also apply the automaton synthesis algorithm for the Fetch environment. The reduced automaton of *PickLiftPlace* task is presented in Figure 4. It is decomposed into 5 states. $q_5$ is a terminate state, which represents that the whole fetch task succeeds. The edges represent the low-level action policies. These policies can be concluded as approach, grab, lift, and reach target position. At $q_3$, we have 2 paths which can lead to the terminate state. This is because when the target state is above the horizon, the lift and reach action can be further combined into one policy, which is the shortcut edge from $q_3$ to $q_5$. From this perspective, HiLoRL also has the ability to generate its own high-level policy instead of executing low-level policy in a sequential arrangement.

$$
\begin{aligned}
q_1 &\leftarrow P_{0,55} \wedge P_{0,56} \wedge P_{0,57} \wedge P_{0,58} \wedge P_{0,59} \wedge P_{0,60} \wedge P_{0,62} \\
q_2 &\leftarrow P_{0,20} \wedge P_{0,16} \wedge P_{0,17} \wedge P_{0,58} \wedge P_{0,59} \wedge P_{0,60} \wedge P_{0,62} \\
q_3 &\leftarrow P_{0,20} \wedge P_{0,16} \wedge P_{0,17} \wedge P_{0,8} \wedge P_{0,9} \wedge P_{0,60} \wedge P_{0,62} \\
q_4 &\leftarrow P_{0,20} \wedge P_{0,16} \wedge P_{0,17} \wedge P_{0,8} \wedge P_{0,9} \wedge P_{0,61} \wedge P_{0,62} \\
q_5 &\leftarrow P_{0,20} \wedge P_{0,16} \wedge P_{0,17} \wedge P_{0,8} \wedge P_{0,9} \wedge P_{0,61} \wedge P_{0,63}
\end{aligned}
\tag{7}
$$

Here we list the focused input predicates mentioned in Equation 7, and they are all contained in Table 9.

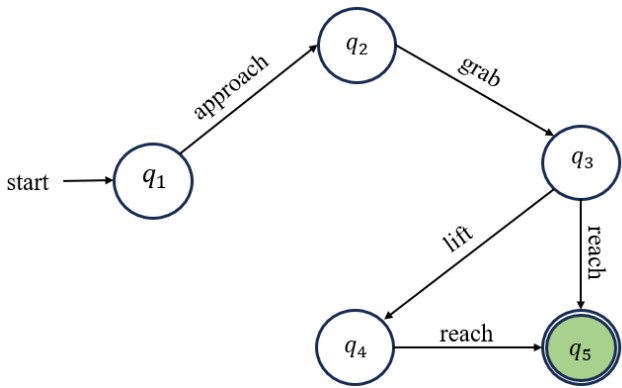

Figure 4: Automaton for PickLiftPlace task in fetch environment.

$P_{0,60}$:   the height of the object is lower than the target height 0.45
$P_{0,61}$:   the height of the object is lower than the target height 0.45
$P_{0,62}$:   the object has not reached the target point
$P_{0,63}$:   the object has reached the target point
$P_{0,55}$:   the x-axis relative distance of the arm and the object is larger than 0.1
$P_{0,20}$:   the x-axis relative distance of the arm and the object is between 0.008 and 0.01
$P_{0,56}$:   the y-axis relative distance of the arm and the object is larger than 0.1
$P_{0,16}$:   the y-axis relative distance of the arm and the object is between 0.006 and 0.008
$P_{0,57}$:   the z-axis relative distance of the arm and the object is larger than 0.1
$P_{0,17}$:   the z-axis relative distance of the arm and the object is between 0.006 and 0.008
$P_{0,58}$:   the displacement of the left claw is larger than 0.1
$P_{0,8}$:   the displacement of the left claw is between 0.002 and 0.004
$P_{0,59}$:   the displacement of the right claw is larger than 0.1
$P_{0,9}$:   the displacement of the right claw is between 0.002 and 0.004

By abstracting our high-level policy into an automaton and extracting the corresponding predicates for each key node, we show the capability of our logic planner to learn more complex logic beyond sequential logic, and the effectiveness and uniqueness of our predicate descriptions of the states.

# E    IMPLEMENTATION DETAILS

All the experiments are carried out on a machine with an Intel Xeon 2.5 GHz processor and 32 GB of RAM, running Ubuntu 22.

## E.1    HIGHWAY ENVIRONMENT

In the highway environment, we have 4 low-level action policies corresponding to *accleration*, *deceleration*, *merge left*, *merge right*. we choose the Deep Deterministic Policy Gradient (DDPG) algorithm for the low-level action policies. We use Adam optimizer to update the parameters in the DDPG model.

The Hyperparameters for the highway environment are shown in Table 6.

## E.2    FETCH ENVIRONMENT

In the Fetch environment, we conduct threeexperimentst Pick&Place, Pick&PlaceCorner, and Pick-LiftPlace. We have 4 low-level action policies corresponding to *approach*, *grab*, *lift*, *reach*. We also choose the Deep Deterministic Policy Gradient (DDPG) algorithm for the low-level action policies. We use Adam optimizer to update the parameters in the DDPG model.

The Hyperparameters for the Fetch environment are shown in Table 7.

| Hyperparameter | Value |
|---|---|
| Pre-training Epoch | 500 |
| Joint Training Epoch | 500 |
| DLM Depth | 7 |
| DLM Breadth | 3 |
| DLM Discount Factor | 0.99 |
| DLM Policy Number | 4 |
| DDPG Discount Factor | 0.99 |
| DDPG Learning Rate | 0.0005 |
| DDPG Replay Buffer Size | 50000 |

Table 6: Hyperparameters in Highway Environment.

| Hyperparameter | Value |
|---|---|
| Pre-training Epoch | 300 |
| Joint Training Epoch | 500 |
| DLM Depth | 3 |
| DLM Breadth | 3 |
| DLM Discount Factor | 0.99 |
| DLM Policy Number | 4 |
| DDPG Discount Factor | 0.95 |
| DDPG Learning Rate | 0.0001 |
| DDPG Replay Buffer Size | 200000 |

Table 7: Hyperparameters in Fetch-Pick-And-Place.

## F  PREDICATES SUMMARY

In this section, we provide the summary of all input predicates and their corresponding relationship with the input states for our two experiments: *Highway* and *Fetch-Pick-And-Place*.

### F.1  INPUT PREDICATES IN HIGHWAY ENVIRONMENT

Here we show the mathematical form of input predicates which is derived from input states in *Highway* Environment. The specific input states and predicates are listed in Table 8.

The meanings of the variables in the input states are as follows:

$d_{x0}$: the x-axis position of the ego agent.
$d_{x1}$: the x-axis position of the nearest car ahead.
$d_{x2}$: the x-axis position of the nearest car behind.
$d_{x3}$: the x-axis position of the nearest car on the left.
$d_{x4}$: the x-axis position of the nearest car on the right.
$d_{y0}$: the y-axis position of the ego agent.
$d_{y1}$: the y-axis position of the nearest car ahead.
$d_{y2}$: the y-axis position of the nearest car behind.
$d_{y2}$: the y-axis position of the nearest car behind.
$d_{y3}$: the y-axis position of the nearest car on the left.
$d_{y4}$: the y-axis position of the nearest car on the right.
$v_{x0}$: the x-axis velocity of the ego agent.
$v_{x1}$: the x-axis velocity of the car ahead.
$v_{x2}$: the x-axis velocity of the car behind.
$v_{x3}$: the x-axis velocity of the car on the left.
$v_{x4}$: the x-axis velocity of the car on the right.
$v_{y0}$: the y-axis velocity of the ego agent.
$v_{y1}$: the x-axis velocity of the car ahead.
$v_{y2}$: the x-axis velocity of the car behind.
$v_{y3}$: the x-axis velocity of the car on the left.
$v_{y4}$: the x-axis velocity of the car on the right.
$e_0$: if there exists a car ahead.
$e_1$: if there exists a car behind.
$e_2$: if there exists a car on the left.
$e_3$: if there exists a car on the right.
$l_0$: the lane in which the ego agent is located.
$l_1$: the target lane.

### F.2  INPUT PREDICATES IN FETCH-PICK-AND-PLACE ENVIRONMENT

Here we show the mathematical form of input predicates which is derived from input states in *Fetch-Pick-And-Place* environment.

We set the activating intervals as follows: $\{0, 0.002, 0.004, 0.006, 0.008, 0.01, 0.012, 0.014, 0.016, 0.018, 0.02, 0.026, 1\}$. They are used to divide the input states into discrete predicates as the input of the high-level logical planner. The specific input states and predicates are listed in Table 9. Except for those predicates, $P_{0,60}$ and $P_{0,61}$ represent if the height of the object is higher than the target height or not based on $z_1$, while $P_{0,62}, P_{0,63}$ represent if the object has reached the target position or not based on $(x_1, y_1, z_1)$.

The meanings of the variables in the Input States are as follows:

$x_0$: The x-axis position of the arm.
$x_1$: The x-axis position of the object.
$y_0$: The y-axis position of the arm.
$y_1$: The y-axis position of the object.
$z_0$: The z-axis position of the arm.
$z_1$: The z-axis position of the object.
$d_0$: The displacement of the left claw.
$d_1$: The displacement of the right claw.

| Activating Intervals | Input States | Predicates | Description |
|---|---|---|---|
| $\{0, 1, 2.5, 5, 10, \infty\}$ | $\|d_{x0} - d_{x1}\|$ | $P_{0,5}, P_{0,6}, P_{0,7}, P_{0,8}, P_{0,9}$ | The x-axis relative distance between the ego agent and the car ahead. |
| $\{0, 1, 2.5, 5, 10, \infty\}$ | $\|d_{x0} - d_{x2}\|$ | $P_{0,10}, P_{0,11}, P_{0,12}, P_{0,13}, P_{0,14}$ | The x-axis relative distance between the ego agent and the car behind. |
| $\{0, 1, 2.5, 5, 10, \infty\}$ | $\|d_{x0} - d_{x3}\|$ | $P_{0,15}, P_{0,16}, P_{0,17}, P_{0,18}, P_{0,19}$ | The x-axis relative distance between the ego agent and the car on the left. |
| $\{0, 1, 2.5, 5, 10, \infty\}$ | $\|d_{x0} - d_{x4}\|$ | $P_{0,20}, P_{0,21}, P_{0,22}, P_{0,23}, P_{0,24}$ | The x-axis relative distance between the ego agent and the car on the right. |
| $\{0, 1, 2.5, 5, 10, \infty\}$ | $\|d_{y0} - d_{y1}\|$ | $P_{0,25}, P_{0,26}, P_{0,27}, P_{0,28}, P_{0,29}$ | The y-axis relative distance between the ego agent and the car ahead. |
| $\{0, 1, 2.5, 5, 10, \infty\}$ | $\|d_{y0} - d_{y2}\|$ | $P_{0,30}, P_{0,31}, P_{0,32}, P_{0,33}, P_{0,34}$ | The y-axis relative distance between the ego agent and the car behind. |
| $\{0, 1, 2.5, 5, 10, \infty\}$ | $\|d_{y0} - d_{y3}\|$ | $P_{0,35}, P_{0,36}, P_{0,37}, P_{0,38}, P_{0,39}$ | The y-axis relative distance between the ego agent and the car on the left. |
| $\{0, 1, 2.5, 5, 10, \infty\}$ | $\|d_{y0} - d_{y4}\|$ | $P_{0,40}, P_{0,41}, P_{0,42}, P_{0,43}, P_{0,44}$ | The y-axis relative distance between the ego agent and the car on the right. |
| $\{0, 0.5, 1, 3, 6, \infty\}$ | $\|v_{x0} - d_{x1}\|$ | $P_{0,45}, P_{0,46}, P_{0,47}, P_{0,48}, P_{0,49}$ | The x-axis relative velocity between the ego agent and the car ahead. |
| $\{0, 0.5, 1, 3, 6, \infty\}$ | $\|v_{x0} - d_{x2}\|$ | $P_{0,50}, P_{0,51}, P_{0,52}, P_{0,53}, P_{0,54}$ | The x-axis relative velocity between the ego agent and the car behind. |
| $\{0, 0.5, 1, 3, 6, \infty\}$ | $\|v_{x0} - d_{x3}\|$ | $P_{0,55}, P_{0,56}, P_{0,57}, P_{0,58}, P_{0,59}$ | The x-axis relative velocity between the ego agent and the car on the left. |
| $\{0, 0.5, 1, 3, 6, \infty\}$ | $\|v_{x0} - d_{x4}\|$ | $P_{0,60}, P_{0,62}, P_{0,62}, P_{0,63}, P_{0,64}$ | The x-axis relative velocity between the ego agent and the car on the right. |
| $\{0, 0.5, 1, 3, 6, \infty\}$ | $\|v_{y0} - d_{y1}\|$ | $P_{0,65}, P_{0,66}, P_{0,67}, P_{0,68}, P_{0,69}$ | The x-axis relative velocity between the ego agent and the car ahead. |
| $\{0, 0.5, 1, 3, 6, \infty\}$ | $\|v_{y0} - d_{y2}\|$ | $P_{0,70}, P_{0,71}, P_{0,72}, P_{0,73}, P_{0,74}$ | The x-axis relative velocity between the ego agent and the car behind. |
| $\{0, 0.5, 1, 3, 6, \infty\}$ | $\|v_{y0} - d_{y3}\|$ | $P_{0,75}, P_{0,76}, P_{0,77}, P_{0,78}, P_{0,79}$ | The x-axis relative velocity between the ego agent and the car on the left. |
| $\{0, 0.5, 1, 3, 6, \infty\}$ | $\|v_{y0} - d_{y4}\|$ | $P_{0,80}, P_{0,81}, P_{0,82}, P_{0,83}, P_{0,84}$ | The x-axis relative velocity between the ego agent and the car on the right. |
| $\{\}$ | $e_i == 1, i = \{0, 1, 2, 3\}$ | $P_{0,1}, P_{0,2}, P_{0,3}, P_{0,4}$ | If there exists a car ahead/behind/on the left/on the right. |
| $\{-\infty, -0.1, 0.1, \infty\}$ | $l_0 - l_1$ | $P_{0,85}, P_{0,86}, P_{0,87}$ | The relative direction between the lane in which the ego agent is located and the target lane. |

Table 8: Input Predicates in Highway Environment.

| Input States | Predicates | Description |
|:---:|:---|:---|
| $\|x_0 - x_1\|$ | $P_{0,0}, P_{0,5}, P_{0,10}, P_{0,15}, P_{0,20}, P_{0,25}, P_{0,30},$ $P_{0,35}, P_{0,40}, P_{0,45}, P_{0,50}, P_{0,55}$ | The x-axis relative distance between the arm and the object. |
| $\|y_0 - y_1\|$ | $P_{0,1}, P_{0,6}, P_{0,11}, P_{0,16}, P_{0,21}, P_{0,26}, P_{0,31},$ $P_{0,36}, P_{0,41}, P_{0,46}, P_{0,51}, P_{0,56}$ | The y-axis relative distance between the arm and the object. |
| $\|z_0 - z_1\|$ | $P_{0,2}, P_{0,7}, P_{0,12}, P_{0,17}, P_{0,22}, P_{0,27}, P_{0,32},$ $P_{0,37}, P_{0,42}, P_{0,47}, P_{0,52}, P_{0,57}$ | The z-axis relative distance between the arm and the object. |
| $\|d_0\|$ | $P_{0,3}, P_{0,8}, P_{0,13}, P_{0,18}, P_{0,23}, P_{0,28}, P_{0,33},$ $P_{0,38}, P_{0,43}, P_{0,48}, P_{0,53}, P_{0,58}$ | The displacement of the left claw. |
| $\|d_1\|$ | $P_{0,4}, P_{0,9}, P_{0,14}, P_{0,19}, P_{0,24}, P_{0,29}, P_{0,34},$ $P_{0,39}, P_{0,44}, P_{0,49}, P_{0,54}, P_{0,59}$ | The displacement of the right claw. |

Table 9: Input Predicates in Fetch-Pick-And-Place Environment.

