# OpenReview forum: "HiLoRL: A Hierarchical Logical Model for Learning Composite Tasks"
_ICLR.cc/2024/Conference — Submitted to ICLR 2024_

### Official Review · Reviewer_732B · 2023-10-14

**Soundness:** 3 good
**Presentation:** 3 good
**Contribution:** 3 good
**Rating:** 6
**Confidence:** 3

**Summary:**

The paper presents a method, named HiLoRL, for reinforcment learning based on adaptive hierarchical logic. HiLoRL consists of a logical planner responsible for high-level decision making and low-level action policies for executing the said planning.  It allows to incorporate expert knowledge in the form of specialized predicates which leads to speed and performances improvements. The authors also present an automation synthesis algorithm to represent the high-level planner, which provides interpretability to their method.

**Strengths:**

I did enjoy reading this paper and I think it has a good amount of novelty and contributions.

* The paper is very well written and presented.
* The approach of decompossing the policy into a high-level logical planer and low-level actions is interesting and as far as I know novel.
* The automaton of the high-level planner provides an interpretable represenation.

I do have some recommendation for the authors in the following.

**Weaknesses:**

On main issue that was not clear is whether the method is applicable to any MDP as a black box or it requires manual design on a per-MDP basis.

On Table 1, it is not easy to compare HiLORL with the baselines as we essentially have a multi-objective criterion, (velocity, crash rate). So it is not accurate to say that PPO and SAC have poor performance. They only have poor performance with respect to vecloity, their crash rate is zero. I recommend the authors re-phrase this or find a fair way to show better performance using a single metric.

On Table 3, it appears that QRM and SPECTRL are not appropriate methods to compare against, given that they have zero success rate. I'd remove those and, ideally, compare against other, more appropriate methods.

The related work section seems to be a bit limited. I'd prefer to see more details about previous work and more importantly a clear connection between HiLoRL and other state of the art work. For instance, in the contributions section at the top of page 2, please mention how HiLoRL compare with other methods.

**Questions:**

Is HiLoRL a human-provided or non-human provided method?

Top of page 3: "this model makes the assumption that an integrated task can be divided into a number of natural and logically interdependent sub-tasks" -> Could you make this assumption more formal and ellaborate a bit?

It is not clear to me how the input module, metioned on page 3, transforms the state into logical predicates. This also appears in line 4 of Alg. 1 and line 3 of Alg. 2. Is there a standarized method to do it for any MDP or do we need to manually design such a module for each MDP?

Appendix A: "possibility distribution" -> "probability distribution"

---

### Official Review · Reviewer_S5sm · 2023-10-20

**Soundness:** 2 fair
**Presentation:** 3 good
**Contribution:** 2 fair
**Rating:** 3
**Confidence:** 4

**Summary:**

This paper presents a hierarchical RL (HRL) framework that features a logical high-level planner selecting among a set of low-level policies to execute. The logical planner is beneficial as: 1) it is naturally interpretable and can be converted into a deterministic automaton, and 2) it can incorporate specialized predicates from domain experts. The logical planner is differentiable and can be trained end-to-end, while the low-level policies are pretrained beforehand and can be jointly tuned together with the logical planner. In experiments, the authors demonstrate the performance of the proposed method against several RL baselines in two environments in OpenAI Gym.

**Strengths:**

The paper is in general clear and easy to follow. The problem of learning logical models as high-level planners for hierarchical RL is interesting, and the authors propose a practical solution where the logical model can be trained end-to-end. The learned high-level planner is interpretable, which is advantageous as the mainstream neural policies are black-box.

**Weaknesses:**

My primary concern with this work is that the proposed method (HiLoRL) fails to demonstrate superior performance, even on the relatively simple control tasks it is evaluated on. Specifically:
1. The control tasks considered are relatively easy to solve because: 1) they are state-based, and 2) a set of pretrained low-level policies are given. I would expect a high-level policy with few layers of MLP trained with PPO can already achieve good performance.
2. In the Highway environment, HiLoRL is compared with PPO and SAC that directly predict low-level actions. This comparison may be unfair - the baselines do not assume a hierarchical structure or leverage the pretrained low-level policies. HiLoRL fails to demonstrate superior performance - it excels in velocity at the cost of a higher crash rate.
3. In the Fetch environment, HiLoRL is not even compared with standard PPO or SAC. How do these methods perform on this benchmark?

In my view, to make a fair comparison, at least some of the baselines need to assume a hierarchical structure and pretrained low-level policies similar to HiLoRL. HiLoRL should at least achieve comparable performance to these baselines before we declare its success.

I’m aware that there usually exists a tradeoff between interpretability and performance of models; however, I believe a logical model is successful only when it can compete with its neural counterparts in performance, while being more generalizable and interpretable.

**Questions:**

1. Following the comments before, how is HiLoRL compared to other HRL methods in performance?
2. I wonder how the training process actually works. It seems that the training of low-level policies and the high-level planner are interdependent. Do you assume the low-level policies are pretrained with a fair quality beforehand?
3. A well-known strength of symbolic/logical models is their capability of compositional generalization. Does HiLoRL have such an ability? For example, generalize to more objects or compositional goals.

---

### Official Review · Reviewer_CNH4 · 2023-10-31

**Soundness:** 2 fair
**Presentation:** 3 good
**Contribution:** 2 fair
**Rating:** 3
**Confidence:** 3

**Summary:**

This work proposes a method to learn a hierarchical model with a high-level planner and low-level policies for composite tasks. The approach can improve its performance by incorporating input expert knowledge (in the form of predicates). The approach combines logic-based representation with classical RL to improve training speed and performance. The empirical evaluation shows that the approach outperforms some baselines in continuous and discrete settings.

**Strengths:**

1. The paper presents an interesting approach for composite tasks.
2. The authors explained the motivation for their work perfectly. It is very clear what problem the paper is addressing.

**Weaknesses:**

### A. *Scope of the work and relationship to previous work:*

1. The novel contribution I see here is the identification of subtasks, as after the subtasks are identified, this approach can be reduced into approaches like [1], which solve a similar problem with the assumption that subtasks (they call them subgoals) are part of the input.
2. Identifying subtasks itself is not a new idea. There are a number of approaches that refer to this problem as identifying skills [2], capabilities [3], landmarks [4], subgoals [1], etc.

Now that we have broken down the proposed approach into these two logical sub-problems, I fail to see the novelty of the problem being solved.

3. The use of logical planner seems like reinventing the wheel as languages like Planning Domain Definition Language (PDDL) already incorporate the first-order logic-based structures proposed in this paper. There are a number of existing planners that can plan at high level using such representations. Consider this specific example: Intensional predicates here can be mapped to axioms in PDDL.




### B. *Weak empirical evaluation:*

1. The empirical evaluation discusses only two domains where the problem horizon does not seem long.  The subtasks in the Fetch environment (Sec. 4.2) seem to be too small. We don’t even know the episode length in this case.
2. I may be wrong here, but I am super confused. In Table 1, how does the crash rate of 0% and longer length for SAC and PPO point to better performance by HiLoRL? Am I missing something? If yes, the explanation needs to be improved.  Or the better figures can be written in bold, like Table 3. Or you can mention things like (higher values better), etc. The text is too verbose in Sec. 4. A rewriting of Sec. 4 will go a long way in improving the paper.
3. Some other unanswered questions from the empirical evaluation. Please refer to Questions 1 to 4 in the “Questions” section.


### C. *Input predicates:*
It is not clear how the input predicates to the DRL module are generated. Subheading “Input module and predicates” in Sec. 3.1 mentioned this in passing. It defines a predicate based on a b-ary transformation function and an activation interval. But who is providing these functions? Is this part of the input? If yes, then can we say predicates are part of the input indirectly? This part is very confusing.
1. Looking at Appendix F, it seems like all the predicates are part of the input. Then, what is the difference between input predicates and expert predicates?
2. If the predicates are input, not learned, then I feel this problem becomes similar to capability discovery [3] or skill learning [2].


### D. *Algorithms:*
Understanding the algorithms is left as an exercise for the reader. There is no text connecting what is happening in the two algorithms with what the overall approach is. This became a major reason for my rejection eventually.



### *Some suggestions to improve the paper* (did not affect my score):
1. I did not see much use in explaining the experimental domains and results in such detail in the main paper when the important parts and crux of the results would’ve sufficed. Instead, the focus can be put more on the approach itself.
2. In sec. 4.1 I did not understand how highway environment is a composite task. This confusion became clear only when I saw Figure 2. You can include a sentence explaining how the task is composite in Sec. 4.1 or move Figure 2 to page 6.
3. There are some places where \citet is used instead of \citep. Please fix it. (E.g., first paragraph of Sec. 3.2.2)

### *Some missing related work* (in addition to the ones mentioned already; did not affect my score)
1. Kokel et al., RePReL: Integrating Relational Planning and Reinforcement Learning for Effective Abstraction. ICAPS 2021.
2. Gehring et al., Reinforcement Learning for Classical Planning: Viewing Heuristics as Dense Reward Generators. ICAPS 2022.

**Questions:**

1. How much do expert predicates improve performance? Is there an experiment that can validate that using expert predicates can improve performance?
2. How are the predicates generated using focused predicate extraction interpretable? Is post-hoc interpretability ended after predicates are extracted? If yes, are there any user studies to support this, similar to [5]?
3. Is the environment fixed across episodes? If not, is the training of low-level policies done for each environment?
4. How do the values reported in the three tables change with increasing the number of episodes?
5. What is human-predictable in line 4 of the abstract?

--------------
### *References*
[1] Lo et al., Goal-Space Planning with Subgoal Models. (https://openreview.net/forum?id=vPS7yxt6oNE).

[2] Silver et al. Learning neuro-symbolic skills for bilevel planning. CoRL 2022.

[3] Verma et al., Discovering User-Interpretable Capabilities of Black-Box Planning Agents. KR 2022.

[4] Mann et al., Approximate Value Iteration with Temporally Extended Actions. JAIR (2015).

[5] Das et al., State2Explanation: Concept-Based Explanations to Benefit Agent Learning and User Understanding. NeurIPS 2023.

---

### Official Review · Reviewer_zuiu · 2023-11-01

**Soundness:** 2 fair
**Presentation:** 2 fair
**Contribution:** 2 fair
**Rating:** 5
**Confidence:** 4

**Summary:**

The paper presents HiLoRL, a Hierarchical Logical Reinforcement Learning approach designed to solve complex tasks by decomposing them into manageable sub-tasks. Unlike methods relying on fixed logical specifications, HiLoRL builds on Differentiable Logic Machines (DLM), an adaptive logical planner, by integrating low-level policies. The model can also integrate expert knowledge via specialized predicates, enhancing training efficiency. The authors show sota performance in tasks with continuous state and action spaces. The approach emphasizes adaptability, interpretability, and the ability to handle logically uncertain tasks.

**Strengths:**

The paper faces hierarchical reinforcement learning (RL) task decomposition by applying DLMs, a neural-symbolic architecture combining learning and logical reasoning.

The proposed model uses DLM to integrate logical planning and RL for a prefixed number of low-level action policies.
Despite the idea of combining symbolic planning rules with RL is not new (e.g. see \cite{illanes2020symbolic}),  the integration of expert knowledge into the model,  as proposed in the paper,  enhances the model's adaptability to real-world scenarios.
In particular,  focused predicates representing the automaton state provide an engaging way of specifying high-level decisions.
The paper provides an interesting contribution to the neural logic field.

Furthermore, integrating expert knowledge into the model is a notable quality, enhancing the model's adaptability to real-world scenarios.

Pseudo-code and visual aids, such as automaton diagrams, help understand and underline the interpretability of the proposed method.

Some experiments show that the model can be compared to current RL methods, such as Proximal Policy Optimization and Soft Actor Critic.

@inproceedings{illanes2020symbolic,
  title={Symbolic plans as high-level instructions for reinforcement learning},
  author={Illanes, Le{\'o}n and Yan, Xi and Icarte, Rodrigo Toro and McIlraith, Sheila A},
  booktitle={Proceedings of the international conference on automated planning and scheduling},
  volume={30},
  pages={540--550},
  year={2020}
}

**Weaknesses:**

The neural symbolic approach proposed is relevant and engaging. However, several aspects still need to be explored, such as computational methods, computational cost, the expressivity of the language, the ability to exploit datasets, the existence of datasets, and supervision requirements, considering that each rule is manually introduced.

The paper has three main weaknesses related to representation,  computation, and discussing limitations.

Representation.\
The paper dedicates a paragraph to first-order logic. However, there is no reason to introduce symbolic logic probabilities or propositional logic since only 0-arity relations are treated, while in first-order- logic predicate symbols are defined for n-arity, with n a positive integer (see e.g. \cite{enderton}, page 68). \
This reduction of logic to simple symbols makes the approach quite confusing since it is hard to understand what is computable and what the assumptions are.

For example, equation (2) treats a set of 'target predicates', and it assumes to compute an index \delta. However, these predicates can be only 0 (False) or 1 (True), and the softmax returns values such that their sum is 1, so it would not select an index unless just a single predicate takes value 1, in which case the index is irrelevant. Yet, this equation is recalled by both algorithms 1 and 2 as a DLM policy. Likewise, both algorithms 1 and 2 have a statement "observe the environment state", what does this mean? How this observation is computed.

Similarly, in the experiment of the highway environment, there is no reference to the number of lanes, roundabouts, intersections, parking, or other properties, which would be tiringly complex to 'axiomatise' by enumerating 0-arity symbols.
While integrating expert knowledge is a strong point, the paper needs to address the challenges associated with designing expert predicates, especially in complex environments.

The paper briefly touches upon low-level policies but needs to provide an in-depth discussion of their significance. Exploring scenarios where randomly initialised low-level policies are used instead of pretrained ones could offer valuable insights into the method's robustness and the importance of the initialisation strategy.

Computation:\
The effort of integrating symbolic reasoning with RL, beyond REINFORCE, can be appreciated; the method proposed does not show the computation expressivity of current approaches to hierarchical RL, such as, for example, \cite{nachum2018data}.\
The paper lacks a detailed analysis of the computational cost associated with HiLoRL, including the training and inference phases. \
Understanding the computational overhead compared to existing methods is essential for practical applications and scalability.

Why not compare with state-of-the-art in neural-symbolic-RL?\
Comparing with other similar approaches would help to make clear the computation, the expressivity of the language, and the overhead of adding rules and introducing an internal metric.

Limitations.\
The paper needs to discuss potential limitations, which would provide a more comprehensive view of the model's capabilities.


%%%%%%%%%%

@article{nachum2018data,\
 title={Data-efficient hierarchical reinforcement learning},\
 author={Nachum, Ofir and Gu, Shixiang Shane and Lee, Honglak and Levine, Sergey},\
 journal={Advances in neural information processing systems},\
 volume={31},\
 year={2018}\
}

@book{enderton2001mathematical,\
title={A mathematical introduction to logic},\
author={Enderton, Herbert B},\
year={2001},\
publisher={Elsevier}\
}

**Questions:**

Please delineate situations where HiLoRL might not be the most suitable choice.

What limitations or trade-offs are associated with incorporating expert knowledge?

Could overly specific predicates limit the model's adaptability to unforeseen scenarios?

How do you handle conflicting predicates from different experts, such as in the highway scenario where safety and velocity might conflict?

---

### Meta-Review · Area_Chair_7keQ · 2023-12-09

**Metareview:**

Synopsis: This paper tackles the problem of learning composite tasks by leveraging symbolic decomposition with a planner. The logical planner leverages predicates as features to abstract states, and takes decisions in an action space that relies on lower-level neural policies as skills. The results show the proposed approach outperforming baseline RL algorithms that do not leverage hierarchical decomposition

Strengths:
+ The paper is well positioned in an active area of research, neurosymbolic RL
+ The ability to incorporate multiple sources of priors, including expert predicates, is useful

Weaknesses:
- The ideas of hierarchical learning, and neurosymbolic learning in RL, are not new. The paper could better highlight what the contributions are in relation to the state of the art.
- The empirical evaluations notable omit comparing to other hierarchical approaches, or neurosymbolic approaches.

**Justification For Why Not Higher Score:**

1. Does not convincingly position the paper w.r.t. existing work
2. Empirical results are insufficient - lacking key baselines.

**Justification For Why Not Lower Score:**

N/A

---

### Decision · Program_Chairs · 2024-01-16

Reject